# Effects of Lacto-Fermented Agricultural By-Products as a Natural Disinfectant against Post-Harvest Diseases of Mango (*Mangifera indica* L.)

**DOI:** 10.3390/plants10020285

**Published:** 2021-02-03

**Authors:** Fernando H. Ranjith, Belal J. Muhialdin, Noor L. Yusof, Nameer K. Mohammed, Muhammad H. Miskandar, Anis Shobirin Meor Hussin

**Affiliations:** 1Department of Food Technology, Faculty of Food Science and Technology, University Putra Malaysia, UPM Serdang, Seri Kembangan 43400, Selangor, Malaysia; hrpf100@gmail.com (F.H.R.); noorliyana@upm.edu.my (N.L.Y.); 2Food Research Unit, Department of Agriculture, Peradeniya 20400, Sri Lanka; 3Department of Food Science, Faculty of Food Science and Technology, University Putra Malaysia, UPM Serdang, Seri Kembangan 43400, Selangor, Malaysia; belal@upm.edu.my; 4Halal Products Research Institute, University Putra Malaysia, UPM Serdang, Seri Kembangan 43400, Selangor, Malaysia; 5Department of Food Science, Faculty of Agriculture, University of Tikrit, Tikrit 34001, Iraq; nameer@tu.edu.iq; 6Technical Advisory Service Unit, Product Development, Research and Advisory Service Division, MPOB, Bandar Baru Bangi, Kajang 43000, Selangor, Malaysia; mhanif@mpob.gov.my

**Keywords:** lactic acid fermentation, agricultural by-products, antifungal activity, mango, postharvest spoilage, quality

## Abstract

Background: the antagonism activity of lactic acid bacteria metabolites has the potential to prevent fungal growth on mango. Methods: the potential of developing natural disinfectant while using watermelon rinds (WR), pineapple (PP), orange peels (OP), palm kernel cake (PKC), and rice bran (RB), via lacto-fermentation was investigated. The obtained lactic acid bacteria (LAB) metabolites were then employed and the in vitro antifungal activity toward five spoilage fungi of mango was tested through liquid and solid systems. Besides, the effect of the produced disinfectant on the fungal growth inhibition and quality of mango was investigated. Results: the strains *Lactobacillus plantarum* ATCC8014 and *Lactobacillus fermentum* ATCC9338 growing in the substrates PKC and PP exhibited significantly higher in vitro antifungal activity against *Colletotrichum gloeosporioides* and *Botryodiplodia theobromae* as compared to other tested LAB strains and substrates. The in-situ results demonstrated that mango samples that were treated with the disinfectant produced from PKC fermented with *L. plantarum* and *L*. *fermentum* had the lowest disease incidence and disease severity index after 16 days shelf life, as well as the lowest conidial concentration. Furthermore, PKC that was fermented by *L. fermentum* highly maintained the quality of the mango. Conclusions: lactic acid fermentation of PKC by *L. fermentum* demonstrated a high potential for use as a natural disinfectant to control *C. gloeosporioides* and *B. theobromae* on mango.

## 1. Introduction

Mango (*Mangifera indica* L.) is one of the most important tropical fruits, due to its high nutrient content, including vitamins, minerals, and phytochemicals. Recently, the mango has become of high economic value in the global fruit market, with an annual export value of more than 1.69 billion US $ [1]. However, fungal diseases are one of the main problems in the mango supply chain, occurring during the production and handling stages. Fungal diseases cause large economic losses, as well as being a health risk for consumers due to the production of mycotoxins. Important post-harvest diseases causing fungi include anthracnose (*Colletotrichum gloeosporioides*), stem-end rot (*Botryodiplodia theobromae, Lasiodiplodia theobromae, Phomopsis mangifera*, or *Dothiorella dominicana*), and black rot (*Aspergillus* sp.). Some fungal infections occur at flowering and colonize endophytically with the protection of the epidermis [2,3]. Therefore, quiescent infections are very complicated to detect and control in the early stages of the life cycle.

Different techniques, such as physical, chemical, and biological methods, have been used to control post-harvest fungal diseases of mango. In the last two decades, chemical methods were very prevalent, due to their low cost and simplicity [4]. However, recent studies have demonstrated the negative impacts of such chemicals, such as high toxicity residuals, genetic mutations of consumers, environmental pollution, and the generation of highly resistant fungal strains [5,6]. The regulations have restricted the use of chemicals, which led to the search for natural alternatives to control fungal growth [7,8]. Bio-preservation is a promising alternative technique that is applied to control the growth of fungi and it is safe for consumers and the environment. Lactic acid bacteria (LAB) and/or their metabolites applied as natural disinfectant have shown promising results, extending the shelf life as well as improving the quality and sensory properties of the crop [9,10]. The ability of LABs to inhibit fungi growth is due to the production of a broad range of primary and secondary antifungal compounds, such as organic acids, fatty acids, and antifungal peptides [11,12,13]. Marín et al. [9] observed that *L. plantarum* significantly reduced the *Botrytis cinerea* incidence of grapes. *L. paracasei* and *L. plantarum* that were isolated from different fruit and vegetables control the growth of *Colletotrichum capsici* in chili peppers [14]. In another study, *L. plantarum* (three strains), *L. paracasei* (four strains), and *L. pentosus* (three strains) isolated from fermented beverages inhibited spore germination by at least 60% and mycelia growth by 100% of *Colletotrichum gloeosporioides*, anthracnose fungi of papaya fruit [10]. Trias et al. [15] also observed that LAB strains, such as FF441, TM128, and PM456, exhibited the significant potential to prevent spoilage fungi in apples, with Sathe et al. [16] reporting that several LAB isolates that were obtained from fresh vegetables could preserve fresh cucumber.

However, the main issue of the lacto-fermentation process is the high cost for the substrate [17], so agricultural by-products are being considered as promising substrates for lacto-fermentation, due to their high nutrient content that can be utilized by LAB [18,19]. Nonetheless, there is limited research regarding the production of natural disinfectants for the post-harvest application of fruits from agricultural by-products via lacto-fermentation. Therefore, this study aimed to screen the antifungal activity of natural disinfectants that were produced from agricultural by-products for post-harvest application of mango fruit. Additionally, the effect of the natural disinfectant on the fungal growth inhibition and quality of mango was determined in situ.

## 2. Results

### 2.1. Antifungal Activity

Five LAB strains in five agricultural by-products were tested for their potential antifungal activity against *Colletotrichum gloeosporioides*, *Botryodiplodia theobromae*, *Aspergillus niger, A. flavus*, and *A. variecolor*. The results indicated that there were significant interaction effects of substrate (S) x LAB strains (L) on antifungal activity (Table 1). Different cell free supernatants (CFS) that are produced from different substrates fermented with selected LAB strains have various levels of antifungal activities, in both the liquid and solid systems. Some of the screened LAB and substrates showed higher or moderate antifungal activity against *C. gloeosporioides, B. theobromae, A. flavus*, and *A. variecolor*. PKC and PP that were fermented with *L. plantarum* ATCC8014 and *L. fermentum* ATCC9338 showed significantly higher antifungal activity than others. However, some of the fermented substrates and LAB strains showed deficient growth inhibition against the selected fungi, especially *A. niger*.

### 2.2. pH Value and Peptide Concentration

Lower pH and higher peptide concentration in CFS are good indicators of higher antifungal ability, due to the increase of organic acids and antifungal peptide concentration as antifungal compounds. The results indicated that there were higher significant interaction effects of LAB x substrate on the pH value as well as the peptide concentration (Table 2). All of the produced CFS samples showed the lower pH and higher peptide concentration values when compared to their substrate samples before fermentation. The pH of different CFS were determined and large differences were observed ranging from 3.19 to 4.39. PKC and PP substrates fermented with *L. plantarum* ATCC8014, *L. fermentum* ATCC9338, and *L. casei* ATCC33 showed the lowest pH values. The total peptide concentration of different CFS was measured and it ranged from 34 to 172 µg mL^−1^ and PKC substrates fermented with *L. plantarum* ATCC8014 and *L. fermentum* ATCC9338 having the highest peptide concentration (172 and 158 µg mL^−1^, respectively).

### 2.3. Number of LAB in the Fermentation Mixtures

Increasing the number of LAB count in fermentation mixture is a reliable indicator of the progress of fermentation. There were significant interaction effects of the only substrate (S) x fermentation time (T) (*p* < 0.001) on the number of LAB count in fermentation mixtures (Table 3). There were no significant effects of other interactions on the number of LAB counts. However, different substrates showed significant differences of LAB count in the progressing of the fermentation process. PKC showed the highest number of LAB count (7.358 log_10_ CFU mL^−1^) after MRS broth as the control, while OP showed the lowest LAB count (6.834 log_10_ CFU mL^−1^) at 72 h after fermentation.

### 2.4. Disease Incidence

Four antifungal CFSs that were produced by *L. plantarum* ATCC 8014 and *L. fermentum* ATCC 9338 in PKC and PP substrates were selected for further experiments due to their higher antifungal activity. The disease incidence (DI) was measured in treated and non-treated mango samples at 25 ± 2 °C during storage. Only the negative control showed a DI value (13.30%) on day 4, and it showed significantly higher DI (*p* < 0.05) values on day 12 and 16 when compared to the treatments and positive control (Figure 1A). After eight days of storage, all of the samples showed a certain level of DI values, but significantly lower than the negative control (*p* < 0.05). In comparison, the DI values of all samples increased in a similar pattern for the treated samples and positive control. On day 12, the negative control reached the peak DI value (100%), although all of the treatments maintained lower DI rates. Furthermore, all of the treatments showed lower or similar DI values when compared to the positive control (100 ppm sodium hypochlorite).

### 2.5. Disease Severity Index

The disease severity index (DSI) of fruit samples was measured on day 16 of storage by measuring the disease affected surface area, demonstrating a significantly high DSI for the positive and negative controls in comparison to the treated samples (Table 4 and Figure 1B). The highest DSI (80%) was observed for the negative control, while the lowest value (13.3%) was observed for the sample that was treated with CFS of PKC fermented by *L. plantarum* ATCC8014. The DSI of all treatments was significantly lower than both the positive and negative controls.

### 2.6. Total Conidial Concentration

The total conidia concentration (log_10_ CFU mL^−1^) was determined in all samples on day 16 after storage at 25 ± 2 °C (Table 4). The highest conidia concentration (6.74 log_10_ CFU mL^−1^) was observed for the negative control sample, with the CFS being produced by PKC fermented with *L. fermentum* ATCC8014 and *L. plantarum* ATTCC9338 exhibiting the lowest total conidia concentrations (5.32 and 5.41 log_10_ CFU mL^−1^) as compared with other treatments as well as controls.

### 2.7. Quality Parameters for Mango

#### 2.7.1. Fruit Weight Loss

Figure 2A shows the cumulative weight loss of mango that is treated with selected CFS with the positive and negative controls during 16 days of storage at 25 ± 2 °C and 85% RH. After 16 storage days, the maximum cumulative weight loss that was observed with the negative control was 23.36%, 13.25% for PKC that is fermented with *L. fermentum* ATCC9338 was the lowest weight loss at day 16. However, the weight loss of all the samples increased with increasing storage time.

#### 2.7.2. Firmness

The initial flesh firmness of all mango samples was 26.49 N, as shown in Figure 2B. Flesh firmness in day 4 storage started to reduce in all samples without significant changes. On day 8, the texture values of PKC that was fermented with *L. plantarum* ATCC8014 (12.42N) and PKC fermented with *L. fermentum* ATCC9338 (12.38N) exhibited significantly higher firmness values when compared with the negative control (6.42N). On day 12 of storage, firmness values were significantly higher for all treatments as well as the positive control when compared with the negative control. On day 16, the texture firmness was 5.00N for PKC that was fermented with *L. plantarum* ATCC8014 and 4.94N for PKC fermented with *L. fermentum* ATCC9338, being significantly higher when compared to the controls.

#### 2.7.3. Total Soluble Solids

The initial total soluble solids (TSS) content of all mango samples was recorded as 9.5 °Brix and started to increase thereafter until day 12 (Figure 2C). The increasing TSS content of the negative control was significant when compared with all the treated samples and the positive control on day 8 and 12. After 12 days of storage, the TSS content of all the treated and control samples was reduced, except mango that was treated by CFS of PP fermented with *L. plantarum* ATCC8014.

#### 2.7.4. Peel Colour

The green colour of all the samples reduced during storage, becoming less green (less negative CIE a∗ value and higher b* value) or near red or yellow colour (Figure 3A,B). Thus, at earlier stage the peel of all mango samples had a moderate green colour (Figure 4), with CIE a∗ and b* values of −15.04 and 30.99, respectively. Four days after storage, the negative control sample rapidly started to change peel colour, with a significant reduction of a* and b* values from day 4 (−10.68, 32.26) to the day 16 (3.55, 39.64). All of the treated samples and the positive control sample maintained the peel green colour when compared to the negative control sample. However, the green colour of all the treated and control samples was reduced in the progressing of storage.

## 3. Discussion

The results of the current study indicate that CFS that is produced from selected LAB strains while using five different agricultural by-products had various levels of antifungal activity, which may be related to the production of different antifungal metabolites, in line with previous studies that reported different LAB strains produced a broad range of primary and secondary antifungal metabolites, especially organic acids and bioactive peptides [20,21,22]. The highest antifungal activities in CFS were produced by *L. plantarum* ATCC8014 and *L. fermentum* ATCC9338 on PP and PKC, which may be related to the production of higher contents of organic acids. The results indicated that the lowest pH values exhibited in CFS were produced by *L. plantarum* ATCC8014 and *L. fermentum* ATCC9338 on PP and PKC. The higher antifungal activity of these samples could be the result of the production of higher levels of organic acids. The acidic environment that is created by the production of organic acids will restrict the growth of fungi [23]. Organic acids directly contribute to the inhibition, as they penetrate the cytoplasm, thereby reducing the cytoplasmic pH causing proton motive force of the target cells [24]. Weak organic acids, such as acetic, lactic, and propionic acid, have a synergistic effect in preventing fungal growth due to their higher pKa value causing them to have a higher level of dissociation inside the host cells [25]. The decreasing pH in the environment results in a greater concentration of protons and increases acid penetration across the cytoplasmic membrane. In response, the microorganism allocates most of its energy to eliminating these newly formed protons, which results in slower growth kinetics. Hence, they become less pathogenic [23]. However, many researchers observed the production of organic acids by LAB strains and their antifungal ability in different levels against various fungi. Gerez et al. [26] reported that acetic acid that is produced by *L. reuteri* 1100 inhibited the growth of *F. graminearum*. Yang and Chang [27] observed that phenyllactic acid that is produced by *L. plantarum* AF1 inhibited the growth of *A. flavus* ATCC 22546. 3-phenyllactic acid and benzene acetic acid produced by *L. plantarum* IMAU10014 inhibited the activities of *Botrytis cinerea, Glomerella cingulate, Phytophthora drechsleri*, and *F. oxysporum* [17].

Bioactive peptides are short chain amino acid structures with low molecular weight, which have great antifungal potential. Liu et al. [28] reported that a novel bio active peptide, CgPep33, inhibited the in vitro growth of *B. cinerea* by 50% at low concentration (20–40 µg mL^−1^) and by 100% at high concentrations (120 µg mL^−1^). The antifungal peptides form small channels in the lipid layer of the cell membrane and promote interactions with target microorganisms and they can also penetrate the cell wall of target fungi [29]. The results of the current study showed that the different CFS produced varied concentrations of peptides, which may contribute to their antifungal activity. The current results showed that lower pH values as well as higher peptide concentrations in the same samples, such as CFS produced by *L. plantarum* ATCC8014 and *L. fermentum* ATCC9338 on PP and PKC. Ricci et al. [30] reported that the production of antifungal metabolites depends on the LAB strains as well as substrates. Ghazvini et al. [31] and Ogunbanwo et al. [32] also confirmed that the production of antifungal compounds and their amount depended on the LAB strains. Pessione and Cirrincione [21] observed that more antifungal peptides are produced by LAB in substrates that are rich in protein and Sanchez-Gonzalez et al. [33] reported higher production of bacteriocins from some LAB strains on polysaccharide substrates. The current results indicated the significant differences of growth kinetics of LAB in different substrates during fermentation. PKC and PP substrates showed the highest number of LAB after MRS broth. These higher numbers of LAB in the mentioned samples may also be related to the production of higher contents of organic acids and antifungal peptides due to the good fermentation process.

The CFS produced by *L. plantarum* ATCC8014 and *L. fermentum* ATCC9338 on PKC and PP substrates had higher antifungal activity (in vitro) against post-harvest fungi of mango, so they were selected for the in situ study, demonstrating that all of the selected antifungal metabolites controlled the disease incidence when applied to mango. Different concentrations of antifungal metabolites, such as organic acids and bioactive peptides on the fruit surface, might reduce the disease incidence in treated mango. The selected CFS reduced disease incidence to a different extent, due to their ability to inhibit mycelia growth and/or conidia germination [16,34,35]. The highest reduction in disease incidence was observed with both CFS that is produced by *L. fermentum* ATCC9338 on PKC and PP, which exhibited higher antifungal activities, as well as lower pH and high peptide concentrations, suggesting that the reduction in disease incidence may be due to the production of antifungal compounds, such as organic acids and bioactive peptides.

Significantly lower disease severity index values were exhibited in all CFS treated samples when compared with both positive and negative controls. The lowest conidia concentrations were observed with CFS samples that were produced by *L. plantarum* ATCC8014 and *L. fermentum* ATCC9338 on PKC, which are the CFS samples demonstrating the lowest disease severity index. Therefore, we can suggest that there was a correlation between the disease severity index and the conidial concentration. Dalié et al. [36] and Gerez et al. [37] reported that the metabolites that are produced by LAB have a great ability to suppress the disease severity through the inhibition of spore germination and mycelia growth of phytopathogenic fungi. Additionally, Svanström et al. [35] described that the organic acids are responsible for the inhibition of conidia production of filamentous fungi. In other studies, Kwak et al. [38], Arulrajah et al. [34], and Liu et al. [39] suggested that bioactive peptides are responsible for the control of conidia production, inhibit conidial germination, induce conidia death, and inhibit the mycelia growth. Taken together, we suggest that the conidia production was inhibited by organic acids and bioactive peptides that are produced by LAB in the present study.

Changes in quality of the mango were also assessed in the present study, by measuring weight loss, peel color change, loss of firmness, and total soluble solids content. The fruit weight loss is a physical process that is caused by the movement of water from the plant tissues to the environment [40]. Many studies have observed that different coating materials control the water loss of fruit and vegetables. Vargas et al. [41] reported that the coatings of strawberries increased the water vapor resistance as compared to the uncoated fruits, as the coating will create a light barrier on the fruit surface. In the present study, treated CFS could behave as a coating material with hydrophobic properties on the mango surface, due to organic acids and antifungal peptides potentially being contained in the CFS [42]. Several studies reported the water vapor barrier properties of the protein, peptides, starch, fatty acids, and organic acids [43,44]. In the present study, the fermented CFS also contains compounds with hydrophobic properties, such as peptide and organic acids. Muhialdin et al. [45] observed that most short chain antifungal peptides have a hydrophobic property that is produced by lacto-fermentation. Hence, we suggest the low weight loss rate of CFS treated mango samples could be due to the water barrier properties of treated CFS. Furthermore, biotic stress due to disease development changes the appropriate cellular responses in the plant tissues [46]. Many researchers have been reported that biotic stress due to disease conditions increases the ethylene production rate of plant tissues, which increases ethylene-induced respiration [47,48]. In the present study, the disease severity index and specific conidial concentration data indicated the size of the fungal population of the mango samples; therefore, it can be suggested that there is an interaction between the fungal population and biotic stress. In a previous study, there was a positive correlation between respiration rate and the weight loss percentage of fruits [49]. In the present study, treated mango samples with lower weight loss values could be due to the low ethylene production, due to low abiotic stress, which reduces the respiration rate and water loss. The highest weight loss of the non-treated negative control could be the result of a high respiration rate with higher abiotic stress due to elevated disease severity index.

Texture and peel colour are vital parameters in determining post-harvest quality as well as consumer acceptance and marketability [50]. The reduction of fruit firmness can be explained as a breakdown of cell wall polysaccharides, including pectin [50,51]. Mango ripening can be described as reduced firmness due to cell wall degradation by different enzymes such as endo-polygalacturonase, exo-polygalacturonase [52] and pectate lyase [53]. Also, the reduction in green peel colour is a physiochemical scenario in combination with the degradation of chlorophylls by chlorophyllase. The internal ethylene concentration is the main factor influencing fruit ripening, such as colour change and softening. Higher ethylene concentrations trigger enzyme activities, including cell wall degradation and chlorophyll degradation enzymes [54]. Furthermore, biotic stress, such as microorganism infections, induce ethylene production [46]. In the present study, fruit firmness and peel green colour of the negative control sample reduced rapidly (Figure 4), which may be associated with higher disease severity (Table 4). In comparison, the samples that were treated with CFS, especially *L. plantarum* ATCC8014 and *L. fermentum* ATCC9338 on PKC, maintained firmness and a green colour. Previous studies reported that phytopathogens, including fungi, could produce cell wall degrading enzymes, such as pectin methylesterases, exo- and endo-polygalacturonases, a variety of endo-glucanases, pectin and pectate lyases, acetyl esterases, and cellulose [55,56,57], which can degrade the cell wall polysaccharides and reduce the fruit firmness [58]. The results of the present study showed the rapid loss of fruit firmness and the green color of negative control samples, which coincided with the highest disease severity index and specific conidia concentration.

The increasing of soluble solids is an important trait of hydrolysis of starch into soluble sugars with fruit maturing and ripening [59] In this study, the increase of soluble solids concentration during storage could be due to the increase of sugars [60]. The higher soluble solid level that was observed with a negative control may be early ripening associated with higher ethylene production that is induced by biotic stress (Figure 4). Furthermore, a slower increase in soluble solids of other samples as compared to the negative control, which may be due to prevent disease development by the treatments. Additionally, the reduction of soluble solids in negative control after day 12 might be early senescence that is associated with higher disease severity and increased titratable acidity level (data not shown) with the hydrolysis of the sugars.

## 4. Materials and Methods

### 4.1. Microorganisms and Culture Conditions

*Lactobacillus plantarum* ATCC8014, *L. fermentum* ATCC9338, *L. casei* ATCC33, and previously isolated *L. plantarum* FC3 and *L. plantarum* Tempeh37 were obtained from the Laboratory of Bioprocessing, Faculty of Food Science and Technology, Universiti Putra Malaysia. The fungal strains used in this study, *Colletotrichum gloeosporioides* DSM62136, *Botryodiplodia theobromae* DSM62078, and *Aspergillus variecolor* DSM3190 were purchased from DSMZ microorganism collection, Braunschweig, Germany, while *Aspergillus niger* and *A. flavus* were obtained from Laboratory of Biotechnology, Faculty of Food Science and Technology, Universiti Putra Malaysia. The LAB cultures were grown in De Man, Rogosa and Sharpe MRS broth (pH 6.5) at 37 °C for 48 h without agitation. After incubation, the LAB strains were washed two times with sterilized distilled water using centrifugation at 4000× *g* for 10 min. under 4 °C (Sigma 3- 18K Sartorius). The LAB cultures were adjusted to 1 × 10^7^ CFU mL^−1^ with peptone water before use. The fungi were grown at 30 °C on potato dextrose agar (PDA) (OXOID, Columbia, USA). The conidia of the fungi were collected from agar plates after seven days, as described by Broekaert et al. [61]. The conidial suspension concentration was estimated while using a hemocytometer (A3835, HEMC, Noida, India) and adjusted to 1 × 10^6^ conidia mL^−1^ [62]. The fungal spore suspensions were stored in 20% glycerol at refrigeration condition.

### 4.2. Substrates Preparation and Lacto-Fermentation Process

The substrates for fermentation were prepared following the method that was described by Erukainure et al. [63], with minor modifications. Watermelon, pineapple, and orange (10 kg; each) were obtained from the local market at Serdang, Selangor, Malaysia. Raw rice bran (RB) was obtained from Faculty of Agriculture, Universiti Putra Malaysia and expeller pressed palm kernel cake (PKC) was obtained from Malaysian Palm oil Board (MPOB), Bandar Baru Bangi, Selangor, Malaysia. The fruit peels were cleaned with tap water in order to remove impurities. The peels were chopped into small pieces in 2 to 3 cm, and then homogenized in distilled water at a ratio of 30:70 (*w/v*) (from preliminary studies) while using simple wet milling. The samples were labeled as watermelon rind (WR), pineapple peel (PP), and orange peel (OP). PKC and RB were sieved and homogenized in distilled water at the same ratio. All of the substrate mixtures (1L) were placed in glass bottles and autoclaved at 121 °C for 15 min., and then allowed to cool at room temperature. The mixtures were inoculated with 2% (*v/v*) of each LAB culture containing 1 × 10^7^ CFU mL^−1^ and incubated at 37 °C for 72 h in anaerobic condition with agitation at 180 rpm (ES20/60 orbital shaker, Biosan, Latvia Republic). The negative controls were 1L of each substrate mixture (unfermented) and they were stored at −40 °C for further analysis. After 72 h fermentation, the mixtures were autoclaved at 121 °C for 15 min. to deactivate the enzymatic and microbial activities. Cell free supernatant (CFS) was prepared through centrifugation at 5000× *g* for 10 min. (Sigma 3-18KS, Germany). The CFS samples were stored at 4 °C until being used for antifungal assays.

### 4.3. Counting of LAB in the Fermentation Mixtures

The growth kinetics of LAB in different fermentation mixtures were measured by counting the number of LAB while using pour plate method with serial dilution technique, as described by Wehr et al. [64]. A dilution series was prepared using 1 mL of fermentation mixture and 1 mL of all dilutions were added to the Petri plates. Thereafter, 15–20 mL of autoclaved MRS agar was placed on the plates after cooling at 40 °C and the plates were mixed gently by circulation motions before solidifying. The plates were placed in anaerobic chambers with AnaeroGen^TM^ atmosphere generation system (Campygen, Oxoid, Basingstoke, U.K.) and they were incubated at 37 °C for 72 h. The plates were made at 0, 24, 48, and 72 h during fermentation as triplicates. Agar plates containing 25–250 LAB colonies were selected, and the colonies were enumerated. The number of colony-forming units per millilitre of fermentation samples were calculated using the following equation:(1)$$\mathrm{Number}\;\mathrm{of}\;\mathrm{LAB}\;\left(\mathrm{CFU}/\;\mathrm{mL}\right)\;=\;\mathrm{Colonies}\;\mathrm{counted}\;\times \;\mathrm{Dilution}\;\mathrm{factor}\;\times \frac{1}{\mathrm{Aliquot}}$$

### 4.4. Antifungal Assays

The antifungal activity of produced CFS samples was determined using two different methods, such as liquid and solid systems, as described by Lavermicocca, Valerio, and Visconti [65], and Ali-Shtayeh and Abu Ghdeib [66], respectively.

#### 4.4.1. Antifungal Assay in Liquid System

The antifungal activity of the CFS was determined by a microassay using sterile 96-well plates. Malt extracts broth (MEB) (100 µL) containing 5 µL of 1 × 10^6^ conidia mL^−1^ suspension was placed into the microwells. Subsequently, 150 µL of CFS was added into the wells as 25 mg mL^−1^ of the final concentration and each treatment was performed in triplicate. The 96-well plates were incubated at 30 °C, and the fungal growth was determined by the OD_600_ while using a microtiter plate reader (Bio-RAD 170-6930, Hercules, California, USA) at 0 h and 72 h. 100 µL MEB containing 5 µL of 1 × 10^6^ conidia mL^−1^ suspension and 150 µL of non-fermented substrates was used as the control. The antifungal activity of the CFS was expressed as the percentage of fungal growth inhibition and calculated using the following equation:(2)$$\mathrm{Inhibition}\;\left(\%\right)\;=\;\frac{\left(72\;h\;\mathrm{control}-0\;h\;\mathrm{control}\right)-\left(72\;h\;\mathrm{treatment}-0\;h\;\mathrm{treatment}\right)}{\left(72\;h\;\mathrm{control}-0\;h\;\mathrm{control}\right)}\times 100$$

#### 4.4.2. Antifungal Assay in a Solid System

The antifungal activity of fermented CFS samples was determined by the poisoned food technique. PDA media amended with 25 mg mL^−1^ concentrations of fermented CFS were autoclaved at 121 °C for 15 min. and then added to labelled Petri plates. After overnight, PDA plates were inoculated with the fungal discs of 5 mm size obtained from seven days old culture. The plates were incubated at 30 °C for eight days, and the fungal colony diameter was measured from day 4 to 8 while using a digital vernier caliper (Mitutoyo, Japan). PDA with unfermented samples were served as the negative controls. The experiment was done in triplicate and the antifungal activity was measured as the inhibition of colony growth while using the following equation:
(3)$$\mathrm{Inhibition}\;\left(\%\right)\;=\;\frac{\mathrm{Colony}\;\mathrm{diameter}\;\mathrm{control}\;\left(\mathrm{mm}\right)-\mathrm{Colony}\;\mathrm{diameter}\;\mathrm{sample}\;\left(\mathrm{mm}\right)}{\mathrm{Colony}\;\mathrm{diameter}\;\mathrm{control}\;\left(\mathrm{mm}\right)}\times 10$$

### 4.5. In-Situ Evaluation

The CFS produced by *L. plantarum* ATCC8014 and *L. fermentum* ATCC9338 on PKC and PP substrates had higher antifungal activity (in vitro) against the pathogenic fungi of mango, so they were selected for the in situ study. Matured green mango (90–100 days after fruit set) variety “Susu” (*Mangifera indica* L.) were obtained from a commercial orchard, Klang, Selangor, Malaysia. The fruit was selected in terms of uniformity in colour, size and freedom from defects. The selected fruits were cleaned with tap water and then subjected to air drying (2 h) before treatments. The prepared fruits were randomly divided into six groups. The fruit groups had different treatments, including group 1 (PKC + *L. plantarum* ATCC8014), group 2 (PKC + *L. fermentum* ATCC9338), group 3 (PP + *L. plantarum* ATCC8014), group 4 (PP + *L. fermentum* ATCC9338), group 5 (sterilized distilled water as the negative control), and group 6 (100 mg L^−1^ sodium hypochlorite as the positive control [67]). 0.2% (*v/v*) tween 20 was added for all CFS samples as a surfactant to improve the adhesion activity [9]. All of the treatments were applied by spraying equally for 30 s on each mango fruit individually [68]. All of the groups were prepared in triplicates and 10 mangoes were included in each replicate. Samples were stored at room temperature (25 ± 2 °C) and humidity (85 ± 3%) for 16 days. The fruit was covered by perforated polyethylene film to control the water losses.

### 4.6. Disease Incidence

Disease incidence (DI) (percentage of diseased fruits) was determined following the method described by Akhtar and Alam [69], while using the following equation. Eight mango fruits were used for each sample in triplicate to assess the DI. The DI was measured for four days intervals during the 16 day storage period.
(4)$$\mathrm{Disease}\;\mathrm{Incidence}\;\left(\%\right)=\frac{\mathrm{Number}\;\mathrm{of}\;\mathrm{diseased}\;\mathrm{fruits}\;\mathrm{in}\;\mathrm{the}\;\mathrm{sample}\;}{\mathrm{Nunmer}\;\mathrm{of}\;\mathrm{total}\;\mathrm{fruits}\;\mathrm{in}\;\mathrm{the}\;\mathrm{sample}}\times 100$$

### 4.7. Disease Severity Index

Disease severity index (DSI) was determined while using the standards for the assessment of fruits diseases proposed by Akhtar and Alam [69]. Eight mango fruits were used for each sample in triplicate in order to determine the DSI. DSI was assessed using the percentage of surface area with disease symptoms of the sample. The diseased area on the fruit surface was determined by visual assessment method while using a scale with five categories such as <20%, 21–40%, 41–60%, 61–80%, and >81%. The assessment of the diseased area was done after 16 days of storage. DSI was determined while using the following equation:(5)$$\mathrm{DSI}\;\left(\%\right)\;=\;\frac{\Sigma \left(a+b+c\ldots \right)}{N\;\times \;Z}\times 100$$

DSI = disease severity indexΣ (a+ b+ c...) = summation score of diseased fruitN = total number of fruit in sampleZ = the score of the highest diseased sample

### 4.8. Specific Conidial Concentration

The samples for conidia count were prepared under the coning and quartering method [70] while using a blended sample by whole peel and pulp of the fruit. One gram of blended fruit sample was diluted with 100 mL of peptone water containing 0.5% of tween 20 in order to prepare the final counting sample. The total number of conidia in the sample were counted using A3835 hemocytometer (HEMC, German) with an optical microscope. The assessment of conidia count was done in triplicate after 16 days of the storage.

### 4.9. Total Peptide Concentration

Total peptide concentration of CFS samples was determined while using the O-phthaldialdehyde assay (OPA) described by Goodno, Swaisgood, and Catignani [71]. The OPA working solution was prepared by mixing 50 mL of 100 mM sodium tetraborate solution, 5 mL of 20% (*wt/wt*) sodium dodecyl sulphate solution, 80 mg of OPA containing 2 mL of absolute methanol, and 200 μL of β-mercaptoethanol. For the microassay, 27 μL of diluted CFS and 205 µL of OPA working solution were added into a microwell. Deionized water with an OPA working solution was used as the control. The plates were incubated at room temperature for 3 min. and the total peptide concentration was measured under 340 nm (Bio-RAD 170-6930, Hercules, California, USA). L-serine solution series with concentration of 10 mM to 0.3125 mM was used for the preparation of a standard curve.

### 4.10. pH Value

The pH was determined by direct potentiometry while using a benchtop digital pH meter (Fisherbrand accumet™ AE150, Leicestershire, UK).

### 4.11. Quality Parameters of Mango

#### 4.11.1. Fruit Firmness

The firmness of the fruit flesh was measured for four-day intervals during the 16 day storage period using a texture analyser (TA-XT2i, Japan) with a 5 mm diameter cylindrical probe. The peel of measuring points of the sample was removed by a sharp knife in order to avoid bruising. The texture analyser was used at testing the speed of 2 mm s ^−1^, pre- test speed of 10 mm s ^−1^, post-test speed of 10 mm s ^−1^, and the penetration depth was 10 mm. The average value of firmness for each fruit was calculated while using three values that were taken at three points on the equal area of two fruits in triplicates.

#### 4.11.2. Total Soluble Solids

The average value of total soluble solids (TSS) concentration was measured every four-day intervals during 16 days on three fresh juice samples taken from a blended whole mango fruit in triplicates. The samples were measured using a handheld digital refractometer (HANNA 9680, Woonsocket, RI, USA).

#### 4.11.3. Weight Loss

The weight of treated and controlled mango fruits was measured every four-day intervals in grams per fruit while using an electronic scale with three decimals in a triplicate. The percentage of weight loss was calculated using the following formula:(6)$$\mathrm{Weight}\;\mathrm{loss}\;\left(\%\right)\;=\;\frac{\mathrm{Initial}\;\mathrm{weight}\;\mathrm{of}\;\mathrm{fruit}\;\left(g\right)-\mathrm{Final}\;\mathrm{weight}\;\mathrm{of}\;\mathrm{fruit}\;\left(g\right)}{\mathrm{Initial}\;\mathrm{weight}\;\mathrm{of}\;\mathrm{fruit}\;\left(g\right)}\times 100$$

#### 4.11.4. Peel Colour

The peel colour change of treated and control samples was measured while using four fruit in triplicate every four days. Each fruit was marked in three different positions, and the colour was measured at the same position. The CIE a∗ and b∗ values were measured while using a digital colorimeter (CR- 400, Konica Minolta, Japan). The mango peel colour was expressed as a∗ value (green to red) and b* value (blue to yellow).

### 4.12. Statistical Analysis

A randomized complete block design (RCBD) with two factors was used for the experiment of in vitro antifungal screening and in situ study. There were three blocks for each treatment. The data were subjected to analysis of variance (ANOVA) while using the Statistical Analysis System (SAS), version 9.4 (SAS Institute Inc., Cary, NC, USA). The means were compared using Tukey multiple comparison test at the significance level of *p* ≤ 0.05.

## 5. Conclusions

Agricultural by-products have the potential to be used as substrates in order to produce fungal growth inhibitors via lacto-fermentation with certain lactic acid bacteria strains. The natural disinfectant produced from palm kernel cake and pineapple peel fermented with the strains *Lactobacillus plantarum* ATCC8014 and *Lactobacillus fermentum* ATCC9338 inhibiting the growth of fungi, causing mango spoilage. Moreover, the disinfectant that was produced using palm kernel cake fermented with *Lactobacillus fermentum* ATCC9338 significantly controlled disease incidence, disease severity, and conidia production of the fungi in situ, maintaining the post-harvest quality of mango via controlling weight loss, peel colour change, total soluble solid content, and reducing fruit firmness. Future studies are recommended to determine the protective effects of lacto-fermented agriculture by-products in different storage conditions and combination with other protection techniques. The bioactive compounds that are responsible for the protection of mango quality should be identified to understand their mechanisms of action.

## Figures and Tables

**Figure 1 plants-10-00285-f001:**
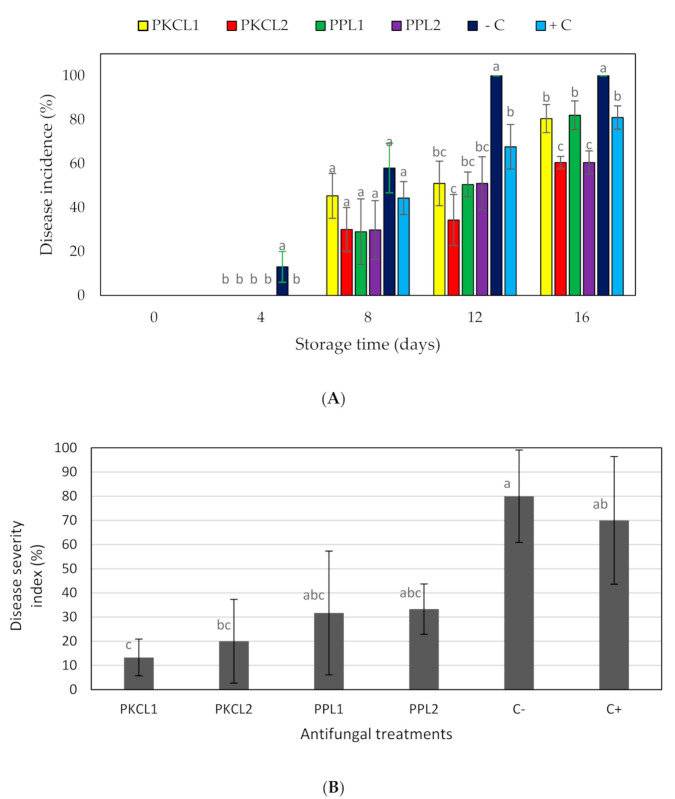
Effect of antifungal treatments on disease incidence (**A**) and disease severity index (**B**) of mango during storage at 25 ± 2 °C for 16 days shelf life. Vertical bars indicate standard error of the mean for three replicate. The sames letter were not significantly different (*p* < 0.05). PKCL1= PKC fermented with *L. plantarum* ATCC8014, PKCL2 = PKC fermented with *L. fermentum* ATCC9338, PPL1 = PP fermented with *L. plantarum* ATCC8014, PPL2 = PP fermented with *L. fermentum* ATCC9338, −C = negative control, and +C = positive control.

**Figure 2 plants-10-00285-f002:**
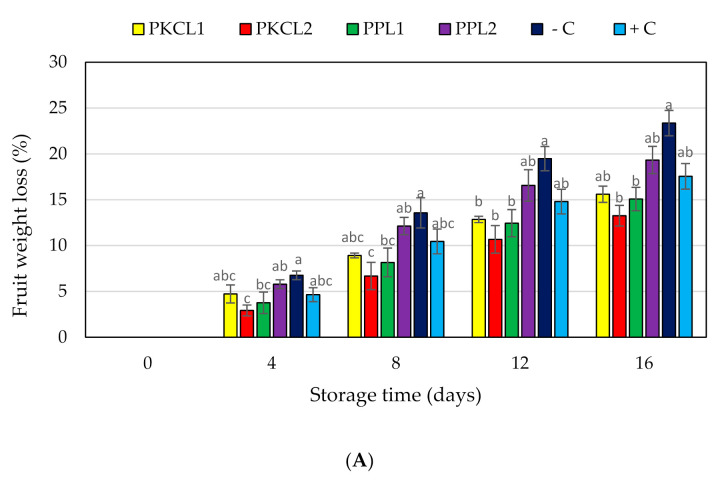
Effect of antifungal treatments and storage days on weight loss (**A**), flesh firmness (**B**) and soluble solids concentration (**C**) of mango fruits during storage at 25 ± 2 °C for 16 days shelf life. Vertical bars indicate standard error of the mean for three replicates. The same letters were not significantly different (*p* < 0.05). PKCL1= PKC fermented with *L. plantarum* ATCC8014, PKCL2 = PKC fermented with *L. fermentum* ATCC9338, PPL1 = PP fermented with *L. plantarum* ATCC8014, PPL2 = PP fermented with *L. fermentum* ATCC9338, −C = negative control, and +C = positive control.

**Figure 3 plants-10-00285-f003:**
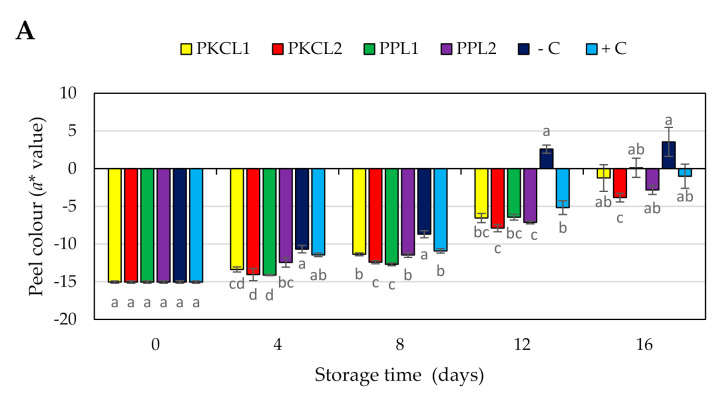
Effect of antifungal treatments and storage days on peel colour, a* value (**A**) b* value (**B**) of mango during Scheme 25 ± 2 °C for 16 days shelf life. Vertical bars indicate standard error of the mean for three replicates. The same letters were not significantly different (*p* < 0.05). PKCL1 = PKC fermented with *L. plantarum* ATCC8014, PKCL2 = PKC fermented with *L. fermentum* ATCC9338, PPL1 = PP fermented with *L. plantarum* ATCC8014, PPL2 = PP fermented with *L. fermentum* ATCC9338, −C = negative control, and +C = positive control.

**Figure 4 plants-10-00285-f004:**
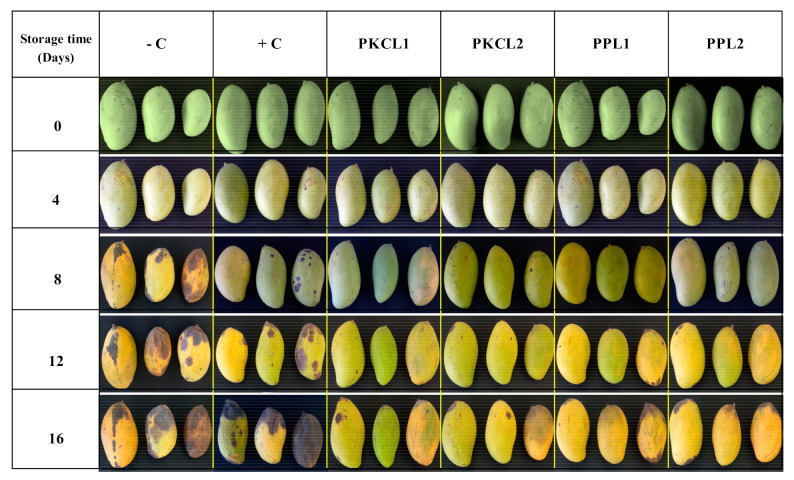
Visual appearance of treated and control mango samples stored at 25 ± 2 °C and 85 ± 3% RH for 16 days shelf life. PKCL1 = PKC fermented with *L. plantarum* ATCC8014, PKCL2 = PKC fermented with *L. fermentum* ATCC9338, PPL1 = PP fermented with *L. plantarum* ATCC8014, PPL2 = PP fermented with *L. fermentum* ATCC9338, −C = negative control, and +C = positive control.

**Table 1 plants-10-00285-t001:** Main and interaction effect of five lactic acid bacteria and five growing substrates on percentage of growth inhibition of five postharvest spoilage fungi in mango by micro well plate assay (liquid system) and agar plate assay (solid system).

Factors	Growth Inhibition (%)				
	*Colletotrichum*	*Botryodiplodia*	*Aspergillus*	*Aspergillus*	*Aspergillus*
	*Gloeosporioides*	*Theobromae*	*Niger*	*Flavus*	*Variecolor*
**Liquid System**					
**LAB strains (L)**					
Control	4.30 ^d^	4.71 ^c^	1.48 ^c^	4.07 ^c^	4.74 ^d^
*L. plantarum* ATCC8014	49.55 ^b^	46.26 ^a^	2.26 ^b,c^	47.02 ^a,b^	34.54 ^b^
*L. fermentum* ATCC9338	62.84 ^a^	50.46 ^a^	11.79 ^a^	44.17 ^b^	40.57 ^a^
*L. casei* ATCC33	38.91 ^c^	43.70 ^a^	2.96 ^b,c^	50.95 ^a,b^	38.34 ^a,b^
*L. plantarum* CF3	38.99 ^c^	33.23 ^b^	8.28 ^a,b^	44.02 ^b^	42.01 ^a^
*L. plantarum* Tempe37	31.64 ^c^	28.67 ^b^	11.81 ^a^	53.80 ^a^	17.00 ^c^
LSD at α 0.05	9.35	8.95	8.63	8.9	4.87
**Substrates (S)**					
WR	28.06 ^b^	32.08 ^b^	3.29 ^b^	37.85 ^b^	18.79 ^d^
OP	24.66 ^b^	20.19 ^c^	5.44 ^a,b^	32.46 ^b^	25.11 ^c^
PP	51.61 ^a^	49.44 ^a^	10.93 ^a^	45.80 ^a^	34.51 ^a,b^
PKC	56.34 ^a^	50.55 ^a^	12.53 ^a^	50.94 ^a^	31.45 ^b^
RB	27.86 ^b^	20.27 ^c^	12.15 ^a^	36.33 ^b^	37.79 ^a^
LSD at α 0.05	8.16	7.8	7.53	7.77	4.25
**Interaction**					
L × S	**	**	**	**	**
**Solid System**					
**LAB strains (L)**					
Control	6.15 ^f^	7.14 ^e^	2.11 ^d^	5.93 ^f^	6.08 ^d^
*L. plantarum* ATCC8014	61.49 ^a^	63.32 ^a^	9.89 ^b^	40.31 ^c^	11.53 ^c^
*L. fermentum* ATCC9338	58.79 ^b^	47.68 ^b^	20.49 ^a^	31.15 ^d^	36.62 ^a^
*L. casei* ATCC33	46.76 ^c^	40.24 ^c^	7.09 ^b^	55.53 ^b^	28.82 ^b^
*L. plantarum* CF3	50.87 ^b^	34.90 ^d^	19.18 ^a^	34.08 ^d^	43.31 ^a^
*L. plantarum* Tempe37	35.05 ^d^	49.25 ^b^	19.47 ^a^	63.32 ^a^	7.07 ^c^
LSD at α 0.05	3.38	3.57	3.17	3.39	4.89
**Substrates (S)**					
WR	35.06 ^c^	53.34 ^b^	13.26 ^b^	56.19 ^b^	13.05 ^c^
OP	32.17 ^c^	40.02 ^c^	34.58 ^a^	29.98 ^d^	33.08 ^a^
PP	68.21 ^a^	52.09 ^b^	7.73 ^d^	62.90 ^a^	24.53 ^b^
PKC	66.61 ^a^	63.17 ^a^	11.70 ^b,c^	49.89 ^c^	16.92 ^c^
RB	50.39 ^b^	26.77 ^d^	8.87 ^c,d^	25.42 ^e^	39.78 ^a^
LSD at α 0.05	3.38	3.59	3.17	3.39	6.89
**Interaction**					
L × S	**	**	**	**	**

Means followed by the same letter within a column are not significantly different at *p* ≤ 0.05 using tukey multiple mean comparison test. ** = highly significant at *p* ≤ 0.01.

**Table 2 plants-10-00285-t002:** Main and interaction effect of five lactic acid bacteria and five growing substrates on pH value and peptide concentration.

Factors	pH	Peptide Concentration (µg mL^−1^)
LAB strains (L)		
*L. plantarum* ATCC8014	3.62 ^e^	113.32 ^b^
*L. fermentum* ATCC9338	3.63 ^e^	117.64 ^a^
*L. casei* ATCC33	3.73 ^d^	97.21 ^c^
*L. plantarum* CF3	3.85 ^c^	93.08 ^d^
*L. plantarum* Tempe37	4.02 ^b^	81.19 ^e^
Sterile substrates	4.88 ^a^	12.75 ^f^
LSD at α 0.05	0.07	2.48
Substrates (S)		
WR	3.85 ^b^	72.93 ^c^
OP	4.26 ^a^	66.49 ^d^
PP	3.63 ^d^	91.43 ^b^
PKC	3.77 ^c^	107.02 ^a^
RB	4.25 ^a^	91.45 ^b^
LSD at α 0.05	0.06	2.16
Interaction		
L × S	**	**

Means followed by the same letter within a column are not significantly different at *p* ≤ 0.05 using tukey multiple mean comparison test. ** = highly significant at *p* ≤ 0.01.

**Table 3 plants-10-00285-t003:** Main and interaction effect of five lactic acid bacteria (LAB) strains, six growing substrates and four fermentation times on LAB growth kinetics during fermentation.

Factors	LAB Counts (log _10_ CFU mL^−1^)
LAB strains (L)	
*L. plantarum* ATCC8014	5.90 ^a^
*L. fermentum* ATCC9338	5.90 ^a^
*L. casei* ATCC33	5.89 ^a^
*L. plantarum* CF3	5.91 ^a^
*L. plantarum* Tempe37	5.90 ^a^
LSD at α 0.05	0.021
Substrates (S)	
MRS broth -control	6.03 ^a^
PKC	5.98 ^b^
PP	5.96 ^b^
OP	5.71 ^e^
WR	5.92 ^c^
RB	5.81 ^d^
LSD at α 0.05	0.023
Fermentation times (T)	
0 h	4.54 ^d^
24 h	5.83 ^c^
48 h	6.05 ^b^
72 h	7.19 ^a^
LSD at α 0.05	0.018
Interaction	
L × S	ns
L × T	ns
S × T	**
L × S × T	ns

Means followed by the same letter within a column are not significantly different at *p* ≤ 0.05 using tukey multiple mean comparison test. Ns = non-significant, and ** = highly significant at *p* ≤ 0.01.

**Table 4 plants-10-00285-t004:** Total conidial concentration and disease severity index of treated and control mango samples after 16 days storage at 25 ± 2 °C.

Treatments	Total Conidial Concentration (log_10_ CFU mL^−1^)	Disease Severity Index (%)
PKCL1	5.41 ± 0.06 ^c^	13.3 ± 7.6 ^c^
PKCL2	5.32 ± 0.03 ^d^	20.0 ± 17.3 ^b,c^
PPL1	5.59 ± 0.02 ^b^	31.7 ± 25.6 ^a,b,c^
PPL2	5.54 ± 0.02 ^b^	33.3 ± 10.4 ^a,b,c^
C−	6.74 ± 0.01 ^a^	80.0 ± 19.1 ^a^
C+	5.54 ± 0.02 ^b^	70.0 ± 26.4 ^a,b^

Means followed by the same letter within a column are not significantly different at *p* ≤ 0.05 using tukey multiple mean comparison test. Data expressed as mean values ± SD. PKCL1 = PKC fermented with *L. plantarum* ATCC8014, PKCL2 = PKC fermented with *L. fermentum* ATCC9338, PPL1 = PP fermented with *L. plantarum* ATCC8014, PPL2 = PP fermented with *L. fermentum* ATCC9338, C− = negative control, and C+ = positive control.

## Data Availability

Not applicable.

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
