# Peer review of "Effects of Lacto-Fermented Agricultural By-Products as a Natural Disinfectant against Post-Harvest Diseases of Mango (Mangifera indica L.)"

_plants, 2021, doi:10.3390/plants10020285_

Round 1
Reviewer 1 Report
General comments:
The authors describe a study where agricultural byproducts where fermented using lactic acid bacteria and the resulting suspension applied to mango in order to control fungal pathogens. The manuscript is well written and the figures and tables describe adequately what was done. The results are impressing. However, I am not sure if there are effects beyond the pH change. Since the authors already knew about the pH effects from the literature, it would have been good to add a control only changing pH. The study is good enough to be published without this control but I would like to suggest adding such a control in future experiments. If you should have performed such a control, please include in the revised version. Additionally it seems the trial was done only once. Our general experience is that fruit quality and disease pressure vary strongly depending on growing year, location and many other factors. Instead of publishing every single experiment may I suggest to aggregate a couple of experiments and produce papers where more factors affecting the fruit/pathogen/control interactions are described. This would significantly reduce reviewer/literature review burden of the whole scientific community without jeopardising quality of research
In order to improve the understandability and the quality of the manuscript I would like to suggest a few minor errors to be changed or a couple of questions to be answered:
Line: 25 higher antifungal activity – compared to what?
Line49: Reference 4 has nothing to do with chemical control methods. What do you mean with genetic mutation being promoted by chemical control methods? Do you mean mutations of the pathogen, the crop or consumers?
Line78: italize species names – throughout the manuscript
Line 78: introduce abbreviation CFS here too
Line 81: …in both liquid….
Table 1: please explain the values. Is this growth inhibition in percentage?
Upper half of both tables are off. Please make sure the lines correspond to the correct strains.
Figure 1: May I suggest to shorten the y-axis to 100% - or take away the label at 120
Line 131: “standard error of the mean”
Line 132: replicates.
The disease severity index shows the most concise and impressive data of your work. Why did you decide to present this “only” as a table. May I suggest doing this as a figure. Maybe even combine with figure 1 as figure 1a and b since its related.
Line212: something wrong with the reference management here ]
Line273: Did you assess organoleptic properties of the mango? Most Mango peel is not eaten but do the fruits smell differently after treatment?
Line279: Why would your CFS be hydrophobic? Please explain
Line282: Although safe to assume that there are some compounds like this in your CFS you did not measure anything. Please reformulate to weaken the statement.
Line 287: I like your discussion of the water loss – its very detailed and informative. However could it be that fungi just increase water loss by wounding fruit – were water vapor from destroyed plant cells evaporate in the environment rather than being consumed by the fungi? I don’t know the answer but maybe you do?
Line288: The water loss seems to correlate well with the spore concentration but not with the severity index. Maybe you find a way to quantify this effect and change the discussion slightly to represent this fact.
Line 291: Many researchers have reported
Line 304: Chlorophyllase
Line 318: The last section of the discussion tries to relate to TSS concentrations while ripening. To me- Line 318 To 324 could be deleted since it speculates o starch degradation and gives results which shouldn’t be in the discussion section anyways. Line 324 to 329 is somewhat relevant to the study but not well described in contrast to the rest of the discussion. May I suggest to either delete the whole section or Start with line 324 and describe a bit better.
Line 318: exhibits
Line319: increase of total
Line 320: This statement is only true if harvested unripe. You describe harvesting at full maturity. To my knowledge, Mangoes at full maturity do not have starch left.
Line 321 increase of
Line 324 what do you mean with “minus control”?
Line339: what is the reason not to use agitation for growth of LAB strains?
Line341: what did you use for adjusting concentration?
Line388: I do not understand the labelling of positive and negative. Are you talking about inhibition? Then a positive control would be something that inhibits the fungi. But is your assumption that the non-fermented substrate inhibits the fungi more than the fermented? If yes, there would be no reason to ferment. If no, then its not a positive control. However, MEB with conidia (I assume the fungi grow on this media) seems to be a proper negative control for inhibition
Line391: equation. Is it the negative or the positive control that you use in this equation?
Line 396: of 7 days old what? Please clarify
Line398 again I would propose to label this as two sets of positive control, but no proper negative control without the addition of mycelium
Line404: How did you define “fully mature”? Did you use brix, acids, firmness, colour values or a mixture thereof? Alternatively, did you use a color chart to assess maturity- please describe. The fact that colour significantly changes over the course of the 16d trial leads to the assumption that they were not fully mature at the start of the experiment
Link: 411: I think I can understand those controls but
Line 420:“assess”
Line 424: are these the same fruits as above or extra?
Line 436. Did you blend the fruit as a whole? Alternatively, peel and pulp separately?
Line 480: is there a special reason that you use Fisher LSD instead of Tukey or Dunnett? From my humble knowledge in statistics I think Fisher LSD doesn’t correct for multiple comparisons which would – given the number of tests you do in this manuscript- be appropriate
Line 483: Is disinfectant the proper word? As you describe it, it is more a fungal growth inhibition than a disinfectant.
Author Response
Review 1:
1# However, I am not sure if there are effects beyond the pH change. Since the authors already knew about the pH effects from the literature, it would have been good to add a control only changing pH. The study is good enough to be published without this control but I would like to suggest adding such a control in future experiments. If you should have performed such a control, please include in the revised version
Reply: Authors added the results of pH and peptide concentration values of sterile substrates as the control as recommended by the respected reviewer. We had enough data about unfermented sterile substrates, such as pH, peptide concentration, etc...(Table 2)
Authors added a new sentence to the ‘pH value and peptide concentration’ in results section, line 102, 103.
2# additionally it seems the trial was done only once. Our general experience is that fruit quality and disease pressure vary strongly depending on growing year, location and many other factors. Instead of publishing every single experiment may I suggest to aggregate a couple of experiments and produce papers where more factors affecting the fruit/pathogen/control interactions are described. This would significantly reduce reviewer/literature review burden of the whole scientific community without jeopardising quality of research
Reply: The in vitro experiments were all carried out in triplicate. Thus, the in-situ trail using the mango fruits was done one time but with high number of replication (16 fruits per group) to be representative. The authors had several limitations to repeat the experiment including 1) the high cost for the mango samples, 2) to avoid wastage of the fruits, 3) the season for mango is very short and the authors could not get the fruits of same maturity. In addition, this is the first study to screen the lacto-fermented PKC as natural disinfectant for mango. Thus, the authors ensured the accuracy for the observation and determination of the fungal growth.
3# 25 higher antifungal activity – compared to what?
Reply: Authors added a part of sentence in abstract to explain to the comparison of antifungal activity as recommended by the respected reviewer, Line 27.
4# Line49: Reference 4 has nothing to do with chemical control methods.
Reply: Authors removed pervious reference [4] and added new reference for most relevant as recommended by the respected reviewer, Line 533-534.
Authors added several words as ‘of consumers’ as recommended by the respected reviewer, Line 53.
5# Line78: italize species names – throughout the manuscript
Reply: Authors corrected all species name as italic fonts including line 82
6# Line 78: introduce abbreviation CFS here too
Reply: Authors added abbreviation for CFS line 84
7# Line 81: …in both liquid….
Reply: Authors corrected, line 83
8# Table 1: please explain the values. Is this growth inhibition in percentage?
Reply: Authors added the explanation before growth inhibition values, line 92.
9# Upper half of both tables are off. Please make sure the lines correspond to the correct strains.
Reply: Authors corrected the tables.
10# Figure 1: May I suggest to shorten the y-axis to 100% - or take away the label at 120.
Reply: Authors shortened the y-axis to 100% as recommended by the respected reviewer, line 142.
11# Line 131: “standard error of the mean”
Reply: Authors corrected the error, lines 146, 187, 222.
12# Line 132: replicates.
Reply: Authors corrected the sentence, line 187.
13# The disease severity index shows the most concise and impressive data of your work. Why did you decide to present this “only” as a table. May I suggest doing this as a figure? Maybe even combine with figure 1 as figure 1a and b since its related.
Reply: Authors added the disease severity index as a new figure (figure 1b) as recommended respected reviewer, line 143.
14# Line273: Did you assess organoleptic properties of the mango? Most Mango peel is not eaten but do the fruits smell differently after treatment?
Reply: The authors totally agree with the respected reviewer, there may be effects on the smell if the natural disinfectant has strong smell. Thus, the selected natural disinfectant from PKC had very light smell. Therefore, the authors did not assess the organoleptic properties of treated mango peel.
15# Line279: Why would your CFS be hydrophobic? Please explain
Reply: Authors explained the statement, line 300.
16# Line282: Although safe to assume that there are some compounds like this in your CFS you did not measure anything. Please reformulate to weaken the statement.
Reply: Authors reformulated the sentence as recommended respected reviewer, line 303.
17# Line 287: I like your discussion of the water loss – it’s very detailed and informative. However could it be that fungi just increase water loss by wounding fruit – were water vapor from destroyed plant cells evaporate in the environment rather than being consumed by the fungi? I don’t know the answer but maybe you do?
Reply: Authors deleted few sentences and rescheduled the paragraph therefore authors couldn’t give enough evidence to confirm the deleted statement, line 306-308.
18# Line 291: Many researchers have reported
Reply: Authors corrected, line 309.
19# Line 304: Chlorophyllase
Reply: Authors corrected, line 328.
20# Line 318: The last section of the discussion tries to relate to TSS concentrations while ripening. To me- Line 318 to 324 could be deleted since it speculates o starch degradation and gives results which shouldn’t be in the discussion section anyways. Line 324 to 329 is somewhat relevant to the study but not well described in contrast to the rest of the discussion. May I suggest to either delete the whole section or Start with line 324 and describe a bit better.
Reply: Authors wrote again the paragraph related soluble solids concentration under the discussion section.
21# Line 318: exhibits
Reply: Authors have rewrite the paragraph
22# Line319: increase of total
Reply: Authors have rewrite the paragraph
23# Line 320: This statement is only true if harvested unripe. You describe harvesting at full maturity. To my knowledge, Mangoes at full maturity do not have starch left.
Reply: Authors changed the statement as recommended respected reviewer, line 342.
24# Line 324 what do you mean with “minus control”?
Reply: It is a typing mistake, authors corrected it as ‘negative control’, line 345.
25# Line339: what is the reason not to use agitation for growth of LAB strains?
Reply: Authors believe the zero agitation is good for the well growth of LAB with information of previous studies. Ex:
“The Lactobacillus species favor a microaerophilic condition or require low oxygen for growth. Therefore, a slow or zero agitation could produce mild aeration for the species”.
https://www.ncbi.nlm.nih.gov/pmc/articles/PMC6651325/
26# Line341: what did you use for adjusting concentration?
Reply: Authors added the adjusting method, line 363.
27# Line388: I do not understand the labelling of positive and negative. Are you talking about inhibition? Then a positive control would be something that inhibits the fungi. But is your assumption that the non-fermented substrate inhibits the fungi more than the fermented? If yes, there would be no reason to ferment. If no, then it’s not a positive control. However, MEB with conidia (I assume the fungi grow on this media) seems to be a proper negative control for inhibition
Reply: Authors corrected the mistake of the positive and negative control. The non-fermented substrate should be the negative control, line 411.
28# Line391: equation. Is it the negative or the positive control that you use in this equation?
Reply: Negative control, non- fermented substrates
29# Line 396: of 7 days old what? Please clarify
Reply: Authors clarified the statement, line 418.
30# Line398 again I would propose to label this as two sets of positive control, but no proper negative control without the addition of mycelium
Reply: Authors corrected the mistake between the positive and negative controls, line 420.
31# Line404: How did you define “fully mature”? Did you use brix, acids, firmness, colour values or a mixture thereof? Alternatively, did you use a color chart to assess maturity- please describe. The fact that colour significantly changes over the course of the 16d trial leads to the assumption that they were not fully mature at the start of the experiment
Reply: Authors used a simple but accurate method such as counting days after fruit set for confirm the maturity of mango. Mango ‘Susu’ has been published as 90-100 days for maturity. Therefore, we used this criteria with orchard records.
32# Line 420:“assess”
Reply: Authors corrected, line 443.
33# Line 424: are these the same fruits as above or extra?
Reply: Same fruits.
34# Line 436. Did you blend the fruit as a whole? Alternatively, peel and pulp separately?
Reply: Authors blended fruits as whole, peel and flesh together
35# Line 480: is there a special reason that you use Fisher LSD instead of Tukey or Dunnett? From my humble knowledge in statistics I think Fisher LSD doesn’t correct for multiple comparisons which would – given the number of tests you do in this manuscript- be appropriate.
Reply: Authors used Fisher LSD for mean comparison specially its higher accuracy and simplicity. However, authors studied deeply the failures of LSD using for multiple comparisons as the recommendation of respected reviewer. Then, authors decided to change the mean separation test. Currently, authors have done mean separation using Tukey multiple mean separation test.
Authors highly respected to reviewer to very important suggestion.
Reviewer 2 Report
This was a well written manuscript. Experiments were conducted with good skill and thoroughness. This is very impressive work. The introduction and discussion were of very high quality with many good references. However, none of the experiments were repeated. I consider this a major problem in getting this work published. Since all of this work is laboratory based, I do not understand why none of the experiments were repeated. The citation numbers were off throughout the manuscript and will need to be corrected. Also, don't forget to use italics for fungal and bacterial names.
Author Response
Review 2:
1# none of the experiments were repeated. I consider this a major problem in getting this work published. Since all of this work is laboratory based, I do not understand why none of the experiments were repeated. The citation numbers were off throughout the manuscript and will need to be corrected.
Reply: The authors highly appreciate the valuable comments from the respected reviewer. The screening experiment was carried out without repeating due to he huge screening work needed to cover all the by-products that was subjected to lacto-fermentation. Thus, the authors carried out the screening in duplicate for the screening. The other experiments related to the in-situ study was carried out in triplicate. The in-situ trail using the mango fruits was done one time but with high number of replication (16 fruits per group) to be representative. The authors had several limitations to repeat the experiment including 1) the high cost for the mango samples, 2) to avoid wastage of the fruits, 3) the season for mango is very short and the authors could not get the fruits of same maturity.
2# Also, don't forget to use italics for fungal and bacterial names.
Reply: Authors kept all fungi and bacteria names in italic fonts as recommended by the respected reviewer.
Reviewer 3 Report
Thank you for the possibility to review this manuscript entitled ,, Effects of lacto-fermented agricultural by-products as a natural disinfectant against post-harvest diseases of mango (Mangifera indica L.)’’. This is an original paper. It covers a curious aspect with potential application character, especially for biotechnology oriented reader. The authors try to explain the antifungal activity of natural disinfectants produced from agricultural by-products for post-harvest application of mango fruit. Abstract and Introduction are responsible clear they contain all the necessary information. Methodology section is short but well and clearly described. Also discussion provides a concise and clear description of a core idea of the manuscript with confrontation of literature of the field. All the tables are adequate and necessary but should be formatted. Additionally the authors suggest possible practical applications of the study, what is undoubtedly a big strength of the manuscript. Please explain why the authors in the individual figures on the charts, despite the error bars, do not present any statistical differences between the samples ????. Section results is clearly presented and it is adequately supported by the evidence adduced. I suggest that the authors add some graphic abstract that will raise the profile of the article and allow the visual effect of the presented results. The conclusions are logically valid and justified by the evidence adduced. Please correct some tiny grammar mistakes e.g. line 29 should be dot L. .
Author Response
Review 3:
1# All the tables are adequate and necessary but should be formatted.
Reply: Authors formatted in correct formal all the tables
2# Please explain why the authors in the individual figures on the charts, despite the error bars, do not present any statistical differences between the samples?
Reply: Authors added letters to indicate significant differences in possible graphs
3# Please correct some tiny grammar mistakes e.g. line 29 should be dot L. . .
Reply: Authors corrected several grammar mistakes pointed as respectable reviewer.
Round 2
Reviewer 2 Report
The work presented in this paper is of high quality. Unfortunately, none of the experiments were repeated. Repeatability is required for publication in most peer-reviewed journals. There are a few spots in the manuscript where corrections are needed (usually things are not italicized). In addition to the minor edits, many of the citations are lacking italics for fungal or bacterial names. Citation numbers that need corrections include: 6, 10, 13, 14, 17, 18, 27, 28, 31, 32, 44, 45, and 63.
Author Response
1# "The work presented in this paper is of high quality. Unfortunately, none of the experiments were repeated. Repeatability is required for publication in most peer-reviewed journals. There are a few spots in the manuscript where corrections are needed (usually things are not italicized). In addition to the minor edits, many of the citations are lacking italics for fungal or bacterial names. Citation numbers that need corrections include: 6, 10, 13, 14, 17, 18, 27, 28, 31, 32, 44, 45, and 63.”
Reply: The authors totally agree with the respected reviewer that is highly recommended to repeat the experiment for the in-situ study. Thus, the authors had the limitation for collecting same variety and same maturity, mango has very short season. On the other hand, the cost to repeat the experiment is very high due to the high number for replications. However, the authors ensured the accuracy for the data via the high number of replications in each group. The authors corrected all mentioned references and several mistakes of other references as pointed by respected reviewer, Reference number 6, 10, 11, 13, 14, 15, 17, 18, 22, 27, 28, 31, 32, 44, 45, 51, 53, 62, 63. The authors extensively revised several minor mistakes such as italicization and LAB strains, Line: 23, 27, 40, 47, 48, 52, 69, 72, 76, 90, 96, 108, 113, 118, 126, 152, 161, 213, 215, 239, 240, 241, 243, 251, 252, 260, 263, 266, 274, 275, 277, 279, 290, 297, 314, 326, 337, 339, 352, 367, 376, 395, 408, 410, 414, 417, 420, 422, 432, 441, 455, 466, 473, 474, 483, 484, 486, 492, 495, 506, and 507.